# Anisotropic energy transfer in crystalline chromophore assemblies

Ritesh Haldar[1], Marius Jakoby[2], Antoine Mazel[3], Qiang Zhang[1,4], Alexander Welle[1], Tawheed Mohamed[1,5], Peter Krolla[1], Wolfgang Wenzel[4], Stéphane Diring[3], Fabrice Odobel[3], Bryce S. Richards[2,6], Ian A. Howard [2,6] & Christof Wöll [1]

An ideal material for photon harvesting must allow control of the exciton diffusion length and directionality. This is necessary in order to guide excitons to a reaction center, where their energy can drive a desired process. To reach this goal both of the following are required; short- and long-range structural order in the material and a detailed understanding of the excitonic transport. Here we present a strategy to realize crystalline chromophore assemblies with bespoke architecture. We demonstrate this approach by assembling anthracene dibenzoic acid chromophore into a highly anisotropic, crystalline structure using a layer-by-layer process. We observe two different types of photoexcited states; one monomer-related, the other excimer-related. By incorporating energy-accepting chromophores in this crystalline assembly at different positions, we demonstrate the highly anisotropic motion of the excimer-related state along the [010] direction of the chromophore assembly. In contrast, this anisotropic effect is inefficient for the monomer-related excited state.

[1] Karlsruhe Institute of Technology (KIT) Institute of Functional Interfaces (IFG), Hermann-von-Helmholtz Platz-1, Eggenstein-Leopoldshafen 76344, Germany. [2] Karlsruhe Institute of Technology (KIT) Institute of Microstructure Technology (IMT), Hermann-von-Helmholtz Platz-1, Eggenstein-Leopoldshafen 76344, Germany. [3] Universitè Lunam, Universitè de Nantes, CNRS, Chimie et Interdisciplinaritè: Synthèse, Analyse, Modèlisation (CEISAM), UMR 6230, 2 rue de la Houssiniere, Nantes cedex 3 44322, France. [4] Karlsruhe Institute of Technology (KIT), Institute of Nanotechnology (INT), Karlsruhe Institute of Technology (KIT), Eggenstein-Leopoldshafen 76344, Germany. [5] Institute of Physics and Technology, International X-ray Optics Lab, National Research Tomsk Polytechnic University (TPU), 30 Lenin ave, Tomsk 634050, Russia. [6] Karlsruhe Institute of Technology (KIT) Light Technology Institute (LTI), Engesserstrasse 13, Karlsruhe 76131, Germany. These authors contributed equally: Ritesh Haldar, Marius Jakoby. Correspondence and requests for materials should be addressed to R.H. (email: ritesh.haldar@kit.edu) or to I.A.H. (email: ian.howard@kit.edu) or to C.Wöl. (email: christof.woell@kit.edu)

The elemental steps in artificial photosynthetic systems are: photon absorption, funneling of the absorbed energy to a reaction center, and utilization of the energy at the reaction center to drive a desired reaction. Optimization of photon absorption has long been the major goal of chromophore design in synthetic molecular chemistry, with great success being achieved with the strategy of extended π-conjugated systems[1]. Nevertheless, optimizing the transfer of energy within chromophore assemblies to the reaction center is equally important. The transport or diffusion of the exciton (the excited-state carrying the energy added to the molecule by the absorption of a photon) between chromophores depends on the architecture of the assembled chromophores[2–5]. Numerous studies have demonstrated that exciton motion in self-assembled organic molecular systems, conjugated polymers, nanostructures and organic–inorganic hybrid structures, critically depends on mesoscale order[6–8]. Particularly, DNA-origami approaches, topochemical polymers or double-walled carbon nanotubes have been demonstrated as highly efficient materials for directional energy transport[9–13]. Achieving directionality of exciton diffusion in isotropic, amorphous assemblies is not possible; instead anisotropic packing with short and long-range order are essential[14–17]. Recently, coordination networks or metal-organic frameworks (MOFs) have emerged as a popular material in the context of modular chromophore organization and tailored material properties[18–22]. The applications for MOFs go far beyond gas storage and separation, and have demonstrated potential in engineering optoelectronic materials[22]. Since these hybrid crystalline frameworks are assembled by combining metal or metal-oxo nodes and organic linkers, the number of possible architectures and topologies is enormous[23]. Therefore, these crystalline assemblies offer possibilities for arranging chromophores in defined geometries that could provide efficient energy harvesting through long-range exciton transport in an energy harvesting system. In addition, the spatial and orientational organization of the chromophores within the framework allows the anisotropy of the exciton motion

to be controlled; this could lead to designs for one-, two-, and three-dimensional artificial photosynthetic systems[7,21].

Since integrating the powder form of MOFs obtained by conventional solvothermal synthetic methods into optoelectronic devices is not straightforward, for the work described here we rely on surface anchored MOFs (SURMOFs)[24]. These monolithic, highly oriented and crystalline MOF thin films are grown on functionalized substrate surfaces using liquid phase epitaxy (LPE)[24]. The thickness of SURMOFs can be adjusted in a straightforward fashion by adjusting the layer-by-layer (lbl) deposition cycles. More importantly, by employing multi-heteroepitaxy, construction of MOF-on-MOF hetero-architectures is feasible[25,26]. A FRET energy donor-type SURMOF can also contain an acceptor linker keeping the crystallinity intact, provided donor and acceptor linkers have similar length[27]. In order to demonstrate the potential of SURMOFs for creating structures showing significant exciton motion with directionality, we rely on Zn-SURMOF-2 structure[28]. The parent structure presented here, Zn-ADB (1) is assembled from 4,4′-(anthracene-9,10-diyl)dibenzoic acid (ADB) linkers and $Zn^{2+[26]}$. The photoluminescence (PL) data recorded for this crystalline chromophoric assembly reveals the presence of two excited-states; a monomer-related state, which decays into a significantly lower energy excimer-related state with a strongly red-shifted PL signal. To unveil the transport properties of these two excited states, we integrate a FRET acceptor ((2,5-bis(butyl)-3,6-bis(4-carboxylicphneyl-4-yl)-2,5-dihydropyrrolo[3,4-c]pyrrole-1,4-dione, or DPP) into the SURMOFs. By using the layer-by-layer growth method we fabricate different types of hetero-multilayers containing pristine Zn-ADB layers with well-defined interfaces to mixed-linker donor–acceptor structures, as depicted in Fig. 1. Comprehensive steady-state and time-resolved PL studies of the two different SURMOF architectures reveal an unprecedented anisotropic, 1D motion of the excimer-related excitons along the crystallographic [010] direction, parallel to the substrate plane. In contrast, the monomer-related state exhibits transport also along

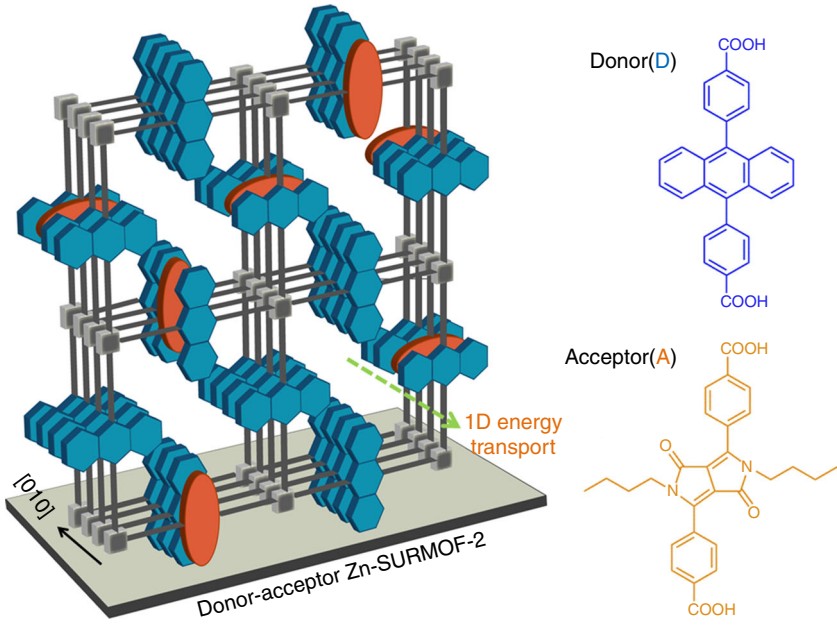

**Fig. 1** Anisotropic architecture of a donor–acceptor SURMOF-2. A schematic of the fabricated Zn-SURMOF-2 structure: (left) a mixed-linker strategy to make mixed-linker donor–acceptor (DA) Zn-SURMOF-2, where the anthracene (blue) chromophores are stacked along [010] direction and the DPP (orange) linkers are homogeneously mixed in the stack. (Right) the individual linkers used to construct the mixed-linker donor–acceptor (DA) SURMOF-2; The blue and orange solids in the SURMOF-2 structure schematic represent ADB and DPP linkers, respectively; gray filled cubes represent the Zn-paddle-wheel secondary building units

Donor(D)

Acceptor(A)

1D energy transport

[010]

Donor-acceptor Zn-SURMOF-2

[100] and [001] direction. The importance of the chromophore organization on the directional long-range FRET hops and diffusion is discussed in detail.

## Results

**Surface grown metal-organic architectures.** The first step in our study of energy harvesting in SURMOFs was to identify a pair of organic linkers that satisfied both of the following criteria: (a) that they shared a similar length so that mixed or heterostructure SURMOFs could be fabricated without any major alteration of the structural order, and (b) exhibited the emission of one of the pair overlapped with the absorption of the other such that the spectral overlap necessary for FRET would be strong. We identified the linker pair ADB as the energy transporting donor, and DPP as the energy-accepting chromophore (Fig. 1; Supplementary Fig. 1). The SURMOFs used here are Zn-SURMOF-2 and were grown on OH-functionalized quartz or silicon substrates using a lbl approach. SURMOF-2 can be viewed as consisting of an array of stacks of square grid type 2D sheets which are formed by connecting paddle-wheel type secondary building units (SBUs) with ditopic carboxylate functionalized organic linkers[28]. Along [010] direction, the linker chromophores can form 1D stacks, as shown in Fig. 1.

We adopted two different approaches to create SURMOF structures in order to characterize the directionality of the exciton motion in the Zn-ADB SURMOF. First, we examine the energy transfer when exciton transport will be dominated by fast hops in the direction of the nearest-neighbor chromophores. In this case, exciton transport will be close to 1D. For this we have used (A) mixed-linker strategy: the donor and acceptor linkers, being similar in length, can be homogeneously mixed and yield a mixed-linker donor–acceptor (DA) crystalline structure, as illustrated in Fig. 1; Supplementary Figs. 1 and 3. To analyze exciton motion along the plane of the 2D sheets, which is anticipated to be slower than along the inter-sheet direction, we have designed a heterostructure using (B) heteroepitaxy method: here the bottom layer is mixed-linker DA SURMOF-2, and a pristine Zn-ADB structure is grown on top (Supplementary Figs. 1 and 3). The out-of-plane X-ray diffraction (XRD) data demonstrate that all the heterostructures grown on quartz or Si described above are highly crystalline, with the [001] direction perpendicular to the substrate. Additional scanning electron microscopy (SEM), time-of-flight secondary ion mass spectrometry (ToF-SIMS) experiments, and ultraviolet-visible (UV-vis) spectrophotometry carried out on these heterostructures are consistent with the structures shown in Fig. 1 (Supplementary Fig. 1–8).

The individual absorption and PL of the linkers and 1 are shown in Supplementary Fig. 9. It is evident that the absorption and PL spectra of 1 differ from those of the ADB linker alone in ethanol. The redshift of the PL spectrum of 1 with respect to the ADB linker PL spectrum in ethanol leads to an excellent overlap with the acceptor (DPP) absorption. Thus, these mixed-linker DA and heterostructure SURMOFs can allow a unique opportunity to carefully analyze the exciton motion.

**Excitonic states in Zn-ADB (1).** Before turning to examine the motion of the excitons in **1** using the mixed-linker DA and heterostructured SURMOFs, we looked into the details of the excited state dynamics in pristine 1. The PL spectrum of 1 ($E_{ex} =$ 3.26 eV) distinctly differs from the monomer state of ADB in ethanol; it is shifted to lower energy compared to that of monomer, and has a broad featureless structure (Supplementary Fig. 9). The time-resolved PL spectra collected with a streak camera system revealed that two distinct excited states are created as a result of photon absorption (Supplementary Fig. 11). The PL shows two components: (a) "monomer-related" feature ($PL_{Mon}$), maximum ~2.81 eV with two sub-nanosecond components (vide infra) and a longer lived tail with a lifetime of 1.2 ns; (b) "excimer-related" feature ($PL_{Exc}$), maximum ~2.58 eV with a lifetime of ~4 ns (for both states compare Supplementary Fig. 11). Figure 2a shows a streak image of the PL of 1 within the first 1500 ps after excitation. The bathochromic shift of the spectrum with longer decay times suggests the presence of at least two different excited states with different lifetimes. In order to determine the decay kinetics of these two states, we used a multivariate fitting scheme[29] that allows the PL spectra and decay profiles of the two different components to be determined (Fig. 2b, c). The first 1.5 ns of the transient $PL_{Mon}$ can be expressed by a biexponential fit with lifetime parameters of $(570 \pm 30)$ (50%) and $(80 \pm 4)$ ps; the $PL_{Exc}$ features a rise time of $(80 \pm 4)$ ps and a decay time of ~4 ns. These observations suggest that a significant fraction of the $PL_{Mon}$ population transfers into the $PL_{Exc}$ state with an inverse rate of 80 ps. Note that this transfer time is substantially longer than the excimer formation time (~150 fs) in the β-form of anthracene crystals[30]. We attribute this divergence to the different type of packing of the anthracene units in 1 and in the bulk anthracene crystals.

To shed some more light on the existence of two excitonic states of 1, we have looked into the possible arrangement of the ADB linkers in SURMOF-2 structure of 1 more carefully. Maintaining the interlayer distance of ~6 Å[26,27], we could optimize the geometry of the neighboring ADB linkers based on optimized potential for liquid simulation (OPLS) force-field (Supplementary Figs. 5 and 6). The optimized geometry

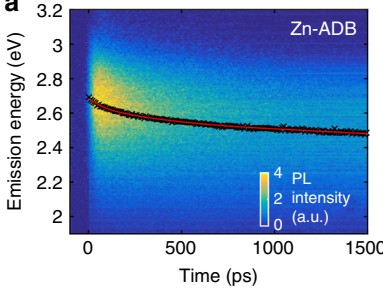
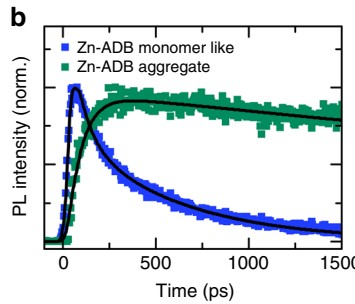
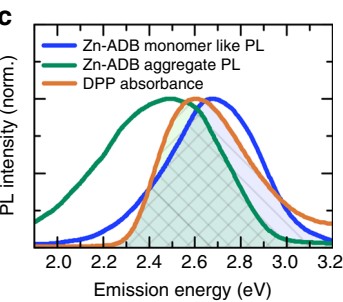

**Fig. 2** Excited states in Zn-ADB (1). **a** Streak image of Zn-ADB (1) excited at 3.26 eV. The peak positions of the PL spectrum at each time (x) are indicated by black dots. The red line shows a spline fit. **b** Decay and rise of the $PL_{Mon}$ and $PL_{Exc}$ states, respectively. The PL transients are fitted by biexponential deconvolution fits (black solid lines). **c** PL spectra of the donor $PL_{Mon}$ and $PL_{Exc}$ states together with the acceptor DPP absorption. Both, the transients and the emission spectra in **b** and **c** have been determined by using a multivariate curve resolution fit[29] with a constrained alternating least squares algorithm

suggests that the closest anthracene-anthracene ring distance is 4.2 Å, which can be regarded as a weak Van der Waals interaction, however not ideal to form an excimer[30–32]. Having the free space available for the rotation of anthracene rings, a 10° rotation of the anthracene ring relative to its optimized position, exhibits a rather small energy change of ~28 kJ/mol and reduces the anthracene-anthracene distance to ~3.4 Å, which is more suitable to form an excimer. This explains why the $PL_{Mon}$ state quickly becomes the $PL_{Exc}$ state and also, vide infra, why a $PL_{Exc}$ state can transfer effectively as other neighboring sites come into similar rotational configurations. The relatively longer formation time of the $PL_{Exc}$ is thus evidently related to rotation of adjacent anthracene unit and the associated geometrical change leading to the excimer formation and reduction of the excited-state energy.

The excited-state population flow can thus be summarized as follow: (1) absorbed photons form $PL_{Mon}$ states, (2) these can decay either to the ground state or to the lower energy $PL_{Exc}$ excited state, (3) the $PL_{Exc}$ state population returns to the ground state (with an inverse rate of 4 ns). We now focus on the diffusion of the two excited states by using DPP as a FRET acceptor. In fact, Fig. 2c demonstrates that DDP is well suited for this purpose since its absorption spectra shows a substantial overlap with both the $PL_{Mon}$ and $PL_{Exc}$ PL spectra.

**Energy transport in mixed-linker DA SURMOF-2.** To examine the energy transfer from the donor 1 to the DPP, we fabricated six different structures as shown in Fig. 1, with increasing amount of DPP concentration from 0.1DPP@1, 0.15DPP@1, 0.45DPP@1, 0.8DPP@1, 2.6DPP@1, and 3.9DPP@1 (where XDPP@1 indicates that the X % of DPP linkers are present in Zn-ADB

SURMOF) (Supplementary Fig. 7). The PL spectra (excited at 3.26 eV) of those donor–acceptor (DA) mixed-linker SURMOF-2 structures demonstrate that with increasing concentrations of DPP, PL of the donor (at 2.75 eV) is strongly quenched, while the acceptor PL (at 2.2 eV) becomes significant (Fig. 3a; Supplementary Fig. 7). Already at DPP concentrations as low as 0.1, most of the donor emission is quenched, as can be seen from Fig. 3a. This observation indicates the presence of a very effective FRET process from 1 to the DPP acceptor in all of these mixed-linker structures. FRET process can also be confirmed by excitation scans monitoring the PL of the DPP acceptor while the excitation wavelength is scanned across the absorption of the donor (1) and the DPP emitter. In all the cases, the maximum emission of the DPP acceptor occurs after excitation of the donor (Supplementary Fig. 13). This confirms that quenching of the donor by FRET to the acceptor is indeed possible in these structures.

To look into the individual quenching efficiencies of the $PL_{Mon}$, and $PL_{Exc}$ states, we have measured the PL decay of the donor, and also PL rise time of the acceptor (Supplementary Fig. 16). The PL decay of the donor at 2.6 eV clearly shows that with increasing concentration of DPP, the donor lifetimes drop continuously; while the DPP acceptor PL rise faster with the increasing concentration of DPP. Extracting the lifetime values of the $PL_{Mon}$, and $PL_{Exc}$ states, we could calculate the quenching efficiencies ($\eta_Q$) as a function of DPP concentration in the mixed-linker structures (Fig. 3b, Supplementary Table 1). The $\eta_Q$ rapidly increases for DPP concentrations under 0.8%, but after 0.8% efficiency stays nearly constant for both the excited states. The fact that the $\eta_Q$ does not change significantly after 0.8DPP@1, is due to a non-uniform distribution of the DPP linkers at higher

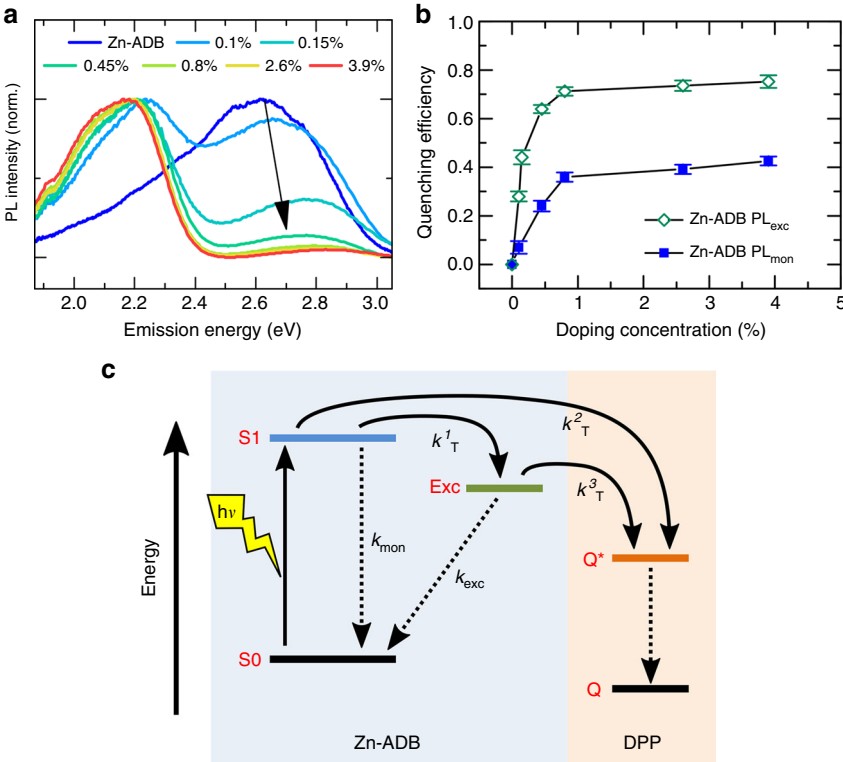

**Fig. 3** Energy transfer in DA SURMOF-2. **a** Normalized PL spectra of the Zn-ADB (1) SURMOF-2 with different concentrations of DPP with excitation at 3.26 eV; with increasing % of DPP, the donor PL decrease. **b** Efficiency of the excited states quenching as a function of the DPP concentration in the mixed-linker DA SURMOF-2 structures calculated from the PL lifetime of the individual excited states (Supplementary Table 1-2). The error bars are determined by propagating the uncertainties in the fits shown in Supplementary Fig. 16. **c** Excited state kinetics of the mixed-linker DA SURMOF-2 structures (also see Supplementary Fig. 15 and Table 1-2)

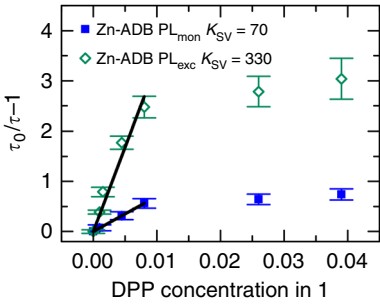

**Fig. 4** Stern–Volmer analyses. Stern–Volmer plot for the DPP containing Zn-ADB (1) SURMOF-2 structures, plotted by comparing the lifetime values of the individual excited states in presence of the DPP in various concentrations. The error bars are determined by propagating the uncertainties in the fits shown in Supplementary Fig. 16

concentrations (see Methods section for details). Approximately, 40% of the $PL_{Mon}$ species and 60% of the $PL_{Exc}$ species are quenched in 0.8DPP@1 (Supplementary Fig. 15). Considering all the involved energy transfer rates, as shown in Fig. 3c, ~76% of the absorbed photons in the end transfer to the DPP acceptor, with half of the energy arriving from a $PL_{Mon}$ state and rest from a $PL_{Exc}$ state (Supplementary Fig. 15 and Table 1-2).

Knowledge of the excited state lifetime as a function of the concentration of the quencher (DPP) allowed us to determine the number of hopping events before the quenching using the Stern–Volmer analysis[33]. Since quenchers are at fixed positions in the lattice each exciton has to hop in average to $K_{sv}$ sites, in order for 50% of the excitons to be quenched. The fit of the linear part of the graph, below 0.8% DPP concentration yields that the $PL_{Mon}$ and $PL_{Exc}$ state visit in average 70 and 330 unique hopping sites, respectively, within their lifetimes (Fig. 4). We note this analysis neglects a final longer range FRET transfer from donor to acceptor, but provides a basis to qualitatively compare the excited sates motion. In order to demonstrate that this diffusion is not isotropic, we have fabricated SURMOFs with a different geometry where quenching can only occur in a bottom layer, and where excitation of the donors is largely confined to the upper layer which only contains ADB but no DPP acceptors.

**Anisotropic energy transport.** In order to demonstrate the presence of directional energy transfer along the [010] direction, parallel to the substrates, we have fabricated SURMOF hetero-multilayers, with a bottom layer consisting of 2.6DPP@1 (~20 nm), and a top layer of the donor 1 with various thicknesses (16, 25, 30, 50, 60, and 90 deposition cycles) (Fig. 5a; Supplementary Fig. 20). Supplementary Fig. 20 shows the PL decay kinetics as a function of donor layer thickness in bilayer heterostructures. A biexponential function fits the decay profiles, and from those the individual excited state's quenching efficiencies can be plotted as a function of donor layer thickness (d) (Supplementary Table 3).

Figure 5b reveals that the quenching efficiency of $PL_{Exc}$ quickly drops when the thickness of the top donor layer is increased. This observation is not consistent with an efficient diffusion along the 2D sheets as revealed by a comparison to the results of a simple simulation considering an isotropic diffusion length of 13 nm and FRET radius (toward DPP) of 5.5 nm, indicated by the broken red line in Fig. 5b (Supplementary Fig. 21). From the clear discrepancy between the results of this simulation and the experimental finding we conclude that the diffusion of the $PL_{Exc}$ state is inefficient along the 2D sheet and essentially confined along the [010] direction, i.e., parallel to the substrate, as shown in Fig. 5a. Indeed, an anisotropic diffusion model where only intralayer FRET hops are allowed (FRET radius of 5.5 nm toward DPP) yields a very good fit of the

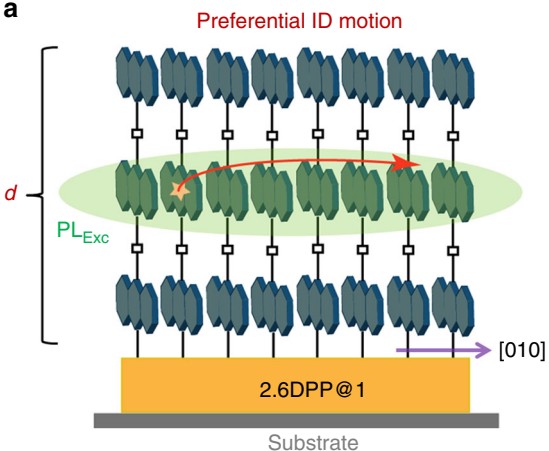

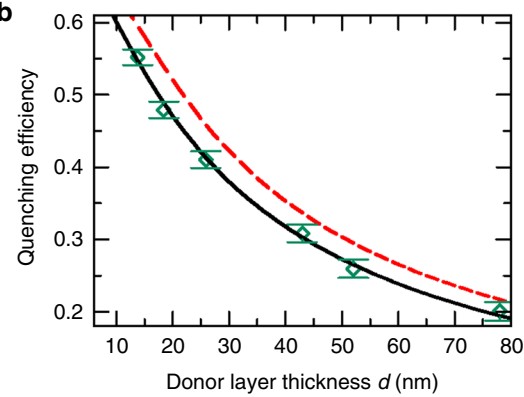

**Fig. 5** Energy transport in bilayer heterostructure. **a** Schematic illustration of the bilayer structure and the anisotropic energy transfer path of $PL_{Exc}$ state. **b** Total quenching efficiencies (shown as green) extracted from the PL decay are shown against the donor layer thickness (d) on top of the 2.6DPP@1; the broken red line is the simulation result assuming isotropic diffusion with FRET hops toward the acceptor and the black solid line is a simulation result considering anisotropic model (only FRET radius of 5.5 nm toward DPP, no diffusion along the sheets) (Supplementary Fig. 19). The error bars are determined by propagating the uncertainties in the fits shown in Supplementary Fig. 20

experimental data, shown as the solid black line in Fig. 5b. Further, a Monte Carlo simulation considering the anisotropic motion of the $PL_{Exc}$ state results a diffusion length of ~97 nm, within its lifetime (Supplementary Fig. 19).

In a sharp contrast, whereas the $PL_{Exc}$ quenching decreases drastically with layer thickness, the quenching of the $PL_{Mon}$ state remains higher. The $PL_{Mon}$ state is more quenched in the bilayer structures than the $PL_{Exc}$. This is in contrast to the mixed-linker DA SURMOFs wherein the $PL_{Exc}$ state showed higher quenching efficiencies than the $PL_{Mon}$ state. Even qualitatively, this provides conclusive evidence that the diffusion of the excimer state is highly anisotropic (with a preferred direction parallel to the substrate), whereas the monomer state diffusion is nearly isotropic (relative to the excimer). The quenching of the monomer state as a function of thickness fit well with a simulation using a diffusion length of 6 nm and FRET radius of 6.5 nm well (Supplementary Fig. 22). Thus, the monomer is able to diffuse in directions, both parallel and perpendicular to the substrate. But the excimer only diffuses parallel to the substrate. The preferential 1D diffusion of the $PL_{Exc}$ state is related to the inter-ADB electronic coupling which promotes a delocalization of the electronic excitation along [010] direction. In contrast, the

$PL_{Mon}$ state diffusion proceeds via nearest-neighbor hopping, and not on delocalized excitonically coupled domains. Hence, the $PL_{Mon}$ state diffuses more efficiently along all crystallographic directions, losing the strongly anisotropic character of the $PL_{Exc}$ state.

## Discussion

The crystalline, 1D anthracene stacks realized via the assembly of a SURMOF represents a type of chromophore organization that allows realizing a directional energy transfer of excited state. Photon absorption by the anthracene cores of the ADB linkers leads to the formation of two excited states, $PL_{Mon}$ and $PL_{Exc}$, which can be distinguished by their different PL energies and lifetimes. By fabricating oriented, highly crystalline SURMOFs, we could demonstrate that the energy transfer for the $PL_{Exc}$ state largely proceeds in the crystallographic [010] direction, thus following the 1D stacks of the anthracene cores of the ADB linkers. In contrast, the $PL_{Mon}$ can also diffuse in the [100] direction perpendicular to the substrate. Similar strong couplings of excitons[34] yielding 1D paths have been realized in purely organic H-type aggregates[8,14]. Our basic theoretical considerations suggest that the 1D motion of $PL_{Exc}$ state has a diffusion length of ~97 nm, however more refined theoretical considerations beyond the point-dipole approximation should increase the accuracy of this finding[7,35].

The ability to fabricate crystalline chromophore assemblies with anisotropic exciton transport creates a huge potential for applications in energy harvesting. In particular, exploiting 1D transport could lead to appreciable diffusion lengths in a desired direction toward an energy accepting layer that could then act as a reaction center performing desired photochemical reactions. Moreover, the directional energy transport in mesoscale crystalline order is hardly achieved, as organic supramolecular designs not often provide such crystalline films. Future efforts will be directed toward the improvement of crystalline order in the SURMOF-based crystalline chromophoric assemblies presented here, in particular to increase the size of the crystalline domains, in order to enhance exciton transport efficiency and diffusion length.

## Methods

**X-ray diffraction (XRD)**. The XRD measurements for out-of-plane (co-planar orientation) were carried out using a Bruker D8-Advance diffractometer equipped with a position sensitive detector Lynxeye in geometry, variable divergence slit and 2.3° Soller-slit was used on the secondary side. The Cu-anodes which utilize the Cu $K\alpha_{1,2}$-radiation ($\lambda = 0.154018$ nm) was used as source.

**Scanning electron microscope (SEM)**. The SEM measurements were carried out using a Field Emission Gun (FEI) Philips XL SERIES 30 ESEM-FEG (FEI Co., Eindhoven, NL). In order to avoid charging and increasing sample conductivity, all samples were coated with a ~5 nm thick gold/palladium film before recording the SEM micrographs. Moreover, condition of High-Vacuum (1.5 Torr) was applied with all specimen, using 20 keV acceleration voltage.

**Atomic force microscope (AFM)**. AFM-imaging was done using an Asylum Research Atomic Force Microscope, MFP-3D BIO. The AFM was operated at 25 °C in an isolated chamber in alternating current mode (AC mode). AFM cantilevers were purchased from Ultrasharp MikroMasch.

**Time-of-flight secondary ion mass spectrometry (ToF-SIMS)**. ToF-SIMS was performed on a TOF.SIMS5 instrument (ION-TOF GmbH, Münster, Germany) equipped with a Bi cluster primary ion source and a reflectron type time-of-flight analyzer. UHV base pressure was $<5 \times 10^{-9}$ mbar. For high mass resolution the Bi source was operated in the "high current bunched" mode providing short $Bi_3^+$ primary ion pulses at 25 keV energy, a lateral resolution of ~4 µm, and a target current of 0.2 pA at 5 kHz repetition rate. The short pulse length of 0.9 ns allowed for high mass resolution. For depth profiling a dual beam analysis was performed in full interlaced mode: The primary ion source was scanned area of $300 \times 300$ µm² ($128 \times 128$ data points) and a sputter gun operated with $Ar_{1650}^+$ cluster ions, 5 keV, scanned over a concentric field of $500 \times 500$ µm², (target current 3 nA) was applied

to erode the sample. Thereby, the sputter ion dose density was >5000 times higher than the Bi ion dose density. Spectra were calibrated on the omnipresent $C^-$, $C_2^-$, $C_3^-$, or on the $C^+$, $CH^+$, $CH_2^+$, and $CH_3^+$ peaks. Based on these datasets the chemical assignments for characteristic fragments were determined. For data visualization, secondary ion intensities were normalized to a maximum of 1.0, each, and plotted over sputter ion fluence [ions/cm²] as an indirect measure for erosion depth.

**Optical characterization**. For the time-resolved spectroscopy, a Hamamatsu Universal Streak Camera C10910 with Acton SpectraPro SP2300 spectrometer, and time correlated single photon counting (TCSPC) with Nano LED light source (373 nm peak wavelength, 1 MHz max. repetition rate, 1.3 ns pulse duration) and FluoroHub Single Photon Detection Module was used. The fluences used in the streak camera system and TCSPC amounted to 160 nJ/cm² and 150 pJ/cm², respectively.

**Fabrication of Zn-ADB (1) SURMOF-2**. Ethanolic solution of 1 mM zinc acetate and 20 µM ADB solutions (in Ethanol) were sequentially deposited onto the substrates using spin coating method in a layer-by-layer fashion. After the metal or linker coating, the samples were rinsed with ethanol to remove unreacted metal/ linker or by-products from the surface. For metal and linker both, the spin coating time is fixed as 10 s with rpm of 2000. The thickness of the samples was controlled by the number of deposition cycles. Forty 40 cycles of deposition resulted ~34 nm film thickness.

**Fabrication of Zn-DPP SURMOF-2 (2)**. Similar to the fabrication of 1, an ethanolic solution of 1 mM zinc acetate and 20 µM DPP solutions (in ethanol) were sequentially spin coated onto the substrates in a layer-by-layer fashion. Forty cycles of deposition resulted ~35 nm film thickness.

**Fabrication of mixed-linker DA SURMOF-2**. The fabrication method is similar to pristine 1; except the linker solution contained 0.1 and 6% of DPP mol/mol. The resulted films however contain 0.1, 0.15, 0.45, 0.8, 2.6, 3.9% of DPP. The UV-vis spectra of those films show a prominent band ~500 nm suggesting the presence of DPP.

**Fabrication of bilayer DA SURMOF-2**. As a bottom layer, 25 cycles of mixed-linker DA SURMOF-2 is grown. On top of that only Zn-ADB SURMOF-2 is grown for various number cycles.

**Geometry optimization of the ADB linkers**. Parameters for ADB ligand are obtained from LigParGen web server[36–38], LigGerGen provides bond, angle, dihedral, and Lennard-Jones OPLS-AA parameters with 1.14*CM1A or 1.14*CM1A-LBCC partial atomic charges. For charged molecules (ADB linker with net charge -2) CM1A charges are NOT scaled by a factor 1.14.

ADB linker is preoptimized based on the OPLS-AA Force Field, duplicate and move the optimized ligand up by 6 Å, freeze (their X, Y, and Z position will not be updated) skeleton carbon atoms (Supplementary Fig. 11) and optimize the dimer in GROMACS.

**Quenching efficiency calculation**. To extract the quenching efficiencies of the top donor layer only we used following equation:

$$Q_t = f_{mix}Q_{mix} + f_{pd}\eta_{p \to mix}Q_{mix}. \qquad (1)$$

Here, $Q_t$ total quenching efficiency ($Q_t$) in the bilayer structure, $f_{mix}$ and $f_{pd}$ are the fractions of the total layer thickness of the 2.6DPP@1 and 1, respectively; $Q_{mix}$ is the quenching efficiencies for the relevant excited state in 2.6DPP@1, $\eta_{p \to mix}$ is the fraction of excitations created in the pristine donor layer (1) that transfer into the bottom layer 2.6DPP@1. Rearranging this formula we solve for $\eta_{p \to mix}$:

$$\eta_{p \to mix} = \frac{Q_t - f_{mix}Q_{mix}}{f_{pd}Q_{mix}}. \qquad (2)$$

Considering the $Q_{mix} \sim 0.7$ (from Fig. 3b) for the $PL_{Exc}$ state, for the 16 and 25 layer thick (corresponding to 6 and 9 ADB linkers, respectively) top layer we find that 32 and 24% of the $PL_{Exc}$ states are transferred to the quenching layer, respectively (Supplementary Fig. 20). While, for the thickest top layer (90 deposition cycles) only 5% of the $PL_{Exc}$ state can reach quenching layer.

In sharp contrast, the $PL_{Mon}$ state diffuses efficiently across the interface to the bottom quenching layer, as can be realized from the transfer efficiencies of 80, 65, and 20% for 14, 20, and 35 nm thick donor layer on top, respectively (Supplementary Fig. 20). To explain such transfer efficiency along the 2D sheet, a diffusion length on the order of ~10 nm (upto 4–5 ADB linkers along the 2D sheet) would be needed. However, this exceeds the diffusion length calculated for purely isotropic diffusion in Supplementary Fig. 19a. Hence, it is likely that the non-

nearest-neighbor FRET from the top layer $PL_{Mon}$ state to the DPP in the mixed-linker layer plays a role in the enhanced transfer rate along the 2D sheets.

**Modeling quenching efficiencies in bilayer structures**. We have modeled the quenching efficiencies in bilayer structures by following Scully et al.[39]. Assuming no gradient of the exciton density in directions parallel to the interface ([010] direction), the quenching efficiencies of the bilayer structures can be modeled by a modified one-dimensional diffusion formula of the following form:

$$\frac{\partial n(x,t)}{\partial t} = D\frac{\partial n(x,t)^2}{\partial x^2} - \frac{n(x,t)}{\tau} - k_F n(x,t) + G(x,t). \quad (3)$$

Here, $n(x,t)$ is the exciton density perpendicular to the donor–acceptor interface and $G(x,t)$ the exciton generation rate at position $x$ and time $t$. The first term on the right side corresponds to the exciton diffusion with a diffusion constant $D$, the second term to the monomolecular decay with a rate $1/\tau$ and the third term to a Förster resonant energy transfer (FRET) to the acceptor with a rate $k_F$. The rate coefficient $k_F$ for FRET is given by:

$$k_F(x) = \frac{c_A}{\tau}\frac{\pi}{6}\frac{R_0^6}{x^3}, \quad (4)$$

with the acceptor density $c_A$ and the FRET radius $R_0$.

At the donor–acceptor interface, we assumed instantaneous quenching of excitons, while at the air film interface reflection no quenching of excitons is assumed. These two boundary conditions are included by the equations: $n(x=0) = 0$, and $\frac{\partial n(x=d)}{\partial x} = 0$, accounting for the quenching and non-quenching interface, respectively.

For all samples under study the absorption length was much bigger than the donor layer thickness $d$ allowing for a uniform exciton density as initial condition. By taking the discussed boundary and initial conditions into account, we have solved the one-dimensional continuity equation (1) numerically for different FRET radii $R_0$, diffusion constants $D$ and layer thicknesses $d$, to gain insights about the diffusions length along sheets of the two excited states in the Zn-ADB SURMOF.

**PL spectra of mixed-linker DA SURMOF-2 structures**. Figure 3a shows the PL spectra of the DPP doped structures after photoexcitation at photon energy of 3.26 eV. We observed that increasing the amount of DPP doping has the following effects: (i) It increases the PL intensity at 2.2 eV (which is related to emission from the DPP acceptors) relative to the PL intensity at 2.75 eV (which is related to the 1). This indicates that, as expected, higher DPP concentrations quench more of the donor emission. (ii) The PL peak of the donor at 2.75 eV shows a blue shift with increasing DPP concentrations. This indicates that the $PL_{Exc}$ states must be mobile and being quenched. If only the $PL_{Mon}$ states were quenched both the $PL_{Mon}$ and $PL_{Exc}$ state PL would decrease by the same amount and there would be no spectral shift of the donor PL. This is because the quenched $PL_{Mon}$ states are also the precursor to the $PL_{Exc}$ state, so the quenching of the $PL_{Mon}$ population would lead to a concurrent decrease in both the PL. The fact that the donor emission blue shifts is therefore clear indication that the $PL_{Exc}$ states are also mobile, participating in energy transport and quenching. Finally, (iii) at higher DPP concentrations the PL peak at 2.2 eV slightly red shifts. This is an indication that at the higher concentrations, the DPP chromophores are not homogeneously distributed but can form small clusters. Interactions between DPP chromophores in these clusters cause the redshifting of the DPP emission, and also effect the average distance that an ADB exciton needs to travel in order to reach a quencher.

## Data availability

The datasets generated during and/or analysed during the current study are available from the corresponding authors on reasonable request.

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

## Acknowledgements

R.H. acknowledges a postdoctoral fellowship from Alexander von Humboldt foundation and M.J. acknowledges support from Karlsruhe School of Optics and Photonics (KSOP) graduate school. A.M., S.D., and F.O. acknowledge Région des Pays de la Loire through the program LUMOMAT for the financial support of this research with the project LumoMOF. ToF-SIMS experiments were supported by the Karlsruhe Nano Micro Facility (KNMF). W.W. acknowledges support from the SFB 1176 "Structuring of Soft Matter".

## Author contributions

R.H. conceived the idea and designed the experiments with M.J., I.A.H., and C.W.; A.M. and S.D. carried out the DPP linker synthesis; R.H. and A.M. carried out the SURMOF syntheses and optimizations; R.H. and M.J. did the photophysical measurements and analyses with help of I.A.H., Q.Z., and W.W. planned the Force Field calculations and Q.Z. carried out the calculations, A.W., T.M., and P.K. helped with material characterizations, R.H., M.J., I.A.H., and C.W. wrote the manuscript with inputs from all the authors.

## Additional information

**Competing interests:** The authors declare no competing interests.

