## [Peer Review File · Nature Communications]

Reviewer #1 (Remarks to the Author):

In the revised version of the manuscript by Haldar et al., reporting on specifically designed surface-anchored metal-organic frameworks for anisotropic exciton diffusion, my previous issues raised have largely been addressed. Yet, two points require further clarification:

1. Novelty and broader interest: As the authors state, it is true that their MOF-structure stands out from existing MOFs in terms of efficient as well as anisotropic energy transfer. Yet, I do not agree with the authors' statement that "the chromophore assembly presented here is distinctly different than the earlier known organic aggregated structures", and that there is a "new type of chromophore assembly". The chromophores in these MOFs form H-aggregates (as evidenced by the spectral features, see Fig. S9), which have been reported for other systems, also for supramolecular assemblies that form by π - π -interactions and/or hydrogen bonding. For the overall photophysics it does not really matter how the structure was formed. Also there is vast amount of reports on excimer formation in supramolecular structures, especially pyrenes are very prone to that. But I guess this is a more general discussion, which is beyond the scope of this paper.

In terms of broader context, in the Conclusions it is stated that MOFs may provide a route to mesoscale crystalline order, which is difficult to achieve for other systems. This is true, but again there is a vast amount of work on organic single crystals (based on small molecules as well as conjugated oligomers and polymers) which have been grown up to many millimetres in size. The authors should report the size of the crystalline domains in their MOFs, and comment on potential routes to increase domain size towards formation of "crystalline films". I find e.g. the statement "... extreme significance towards further development ..." too weak, this should be extended to further highlight the broader interest that this work clearly shows.

2. Exciton annihilation (my previous point 2): Although the fluences used in this work are clearly below those, for which annihilation is typically observed. However, the system presented here is clearly not a "typical" system, it is rather optimised for efficient (Figs. 2 & 4) anisotropic energy transport. As such I believe that annihilation sets in already for fluences significantly below "typical" values, i.e., I would like to see fluence-dependent data to verify the absence of annihilation in the present system. In fact, the observation of a significantly lower annihilation threshold would make this work even stronger. Finally, the fluence reported in the authors' reply should be reported in the paper (I could not find this number).

3. A minor point: Please provide the photon energies used for excitation in Figs. S10, S11, and S14.

After addressing these issues, this very nice manuscript is to my opinion suited for publication in Nature Communications.

Reviewer #2 (Remarks to the Author):

This manuscript reports the different migration behaviors of monomer and excimer excited states in the SURMOFs systems. Through time-resolved PL measurements and simulations, the authors concluded that the excited state of the excimer transport is anisotropic, but the monomer excited state transport is less anisotropic. The authors revised their previous submission and responded to comments from previous reviewers, but there are still issues left unaddressed, rendering it unready for publication.

1. SEM images of these samples should be given. The AFM images (Figure S2) showed the roughness of 5 nm for a thin layer of 35 nm in thickness. The sample surface is not smooth at all after comparing the two numbers. Considering the thickness of some samples to be below 20nm, the roughness of 5nm will be considerable and can greatly affect the photophysical measurement. The authors cannot just “trust” literature precedence, as the scientific conclusion of this paper relies on the correct picture more heavily than those in the previous reports.

2. The monomer excited state cannot be 100% isotropic in its transport in an anisotropic structure. The authors should be careful to not use the word isotropic. I think the authors just want to compare the monomer vs. excimer state. If the authors focus on this comparison to even reflect it in the title, the whole paper will be easier to understand. I strongly suggest the authors to completely rewrite the introduction to focus on the directional transport of excimer excited state vs. monomer excited state.

3. The difference between the isotropic vs anisotropic model prediction in Figure 4b (black vs. red lines) are very small, while the experimental error can be larger than the difference. This renders the whole discussion and conclusion of this paper unconvincing. I am against the publication of this paper because of this point.

4. I do not know what normalization method is used for Figure S13, but it seems to show that energy transfer is not present, since the acceptor emission intensity does not increase at donor absorption region but only increases at acceptor absorption region when the doping level goes up.

5. In the Monte Carlo simulations (Figure S19), the migration of monomer excited state seems also to be anisotropic, why is it not observed in the experiment?

6. A key message the paper should but failed to convey is the reason why excimer excited state transport is more anisotropic than monomer excited state transport. The authors have performed simulations but did not extract the key message from these simulations. The general readers want to know the principles behind it.

Reviewer #3 (Remarks to the Author):

Haldar et al. present an interesting study on a SURMOF designed to produce anisotropic energy transfer. The experiments appear to be well performed with a good set of controls. The data is well presented, and the interpretation of the results seems consistent. Some cross-sectional SEM of the structures would have been good to have, but this is a minor point. I've also been through all the comments from the reviewers at Nature Materials and the authors response and I think they have done a good job in answering the technical questions raised.

Where I do have reservations about this work is the context in terms of light harvesting. The authors pitch is that their system is very good at anisotropic energy transport. But as reviewer 2 has pointed out, this system is clearly not the only such system out there, nor is it close to being the best. The authors response to reviewer 2 on this point is clearly weak.

Many other systems have been developed over the years that show similar features. H-Aggs, J-Aggs and polymer aggregate systems give 1D transport and diffusion lengths of 100nm-4 μ m. To give just a few examples - ref 13, (Nano Lett., 2011, 11 (2), pp 488–492), (Science, Vol. 360, Issue 6391, pp. 897-900, 2018), (J. Phys. Chem. A, 2011, 115 (5), pp 648–654). Many of these systems are easier to process and show longer diffusion lengths. The authors system also involves the motion of the

monomer exciton, which is not anisotropic. From a device point of view this is not desirable, as its essentially a loss channel.

So overall, while this is a very nice experimental study, I think it will mostly be of interest to sections of the MOF community and is unlikely to have a large impact on the larger community working on light harvesting for energy and sensing applications.

Point-by-point response

Reviewer #1 (Remarks to the Author):

In the revised version of the manuscript by Halder et al., reporting on specifically designed surface-anchored metal-organic frameworks for anisotropic exciton diffusion, my previous issues raised have largely been addressed. Yet, two points require further clarification:

We are happy to learn that our answers to the reviewer's comments were mostly satisfactory.

1. Novelty and broader interest: As the authors state, it is true that their MOF-structure stands out from existing MOFs in terms of efficient as well as anisotropic energy transfer. Yet, I do not agree with the authors' statement that "the chromophore assembly presented here is distinctly different than the earlier known organic aggregated structures", and that there is a "new type of chromophore assembly". The chromophores in this MOFs form H-aggregates (as evidenced by the spectral features, see Fig. S9), which have been reported for other systems, also for supramolecular assemblies that form by π - π -interactions and/or hydrogen bonding. For the overall photophysics it does not really matter how the structure was formed. Also there is vast amount on reports on excimer formation in supramolecular structures, especially pyrenes are very prone to that. But I guess this a more general discussion, which is beyond the scope of this paper.

Response: We agree with the reviewer in that numerous H-aggregates or excimers are well studied. It is also correct that in some cases they were aggregated into supramolecular structures. However, we feel that our claim of "new type of chromophore assembly" is justified, for the reason provided below:

- i) In our case, the chromophores form a crystalline structure with well-defined structural parameters. Furthermore, the arrangement of the chromophores leaves a substantial amount of open space (pores within the MOF), which allows for rotational flexibility in the periodic structure. To the best of our knowledge, H-aggregates with these properties have not been described in any work by others.
- ii) in the H-aggregate, we observe two different excited states, which exhibit quite different properties. The first, an excimer-state, shows an unusual, highly anisotropic motion, while the monomer-state exhibits transport along other directions, too.

The presence of these photophysical features make the chromophore assembly unique and shows a pronounced difference to other known supramolecular systems of H-aggregates. We are not

aware of any other works where such anisotropic exciton transport features in a H-aggregate has been observed and carefully characterized.

In terms of broader context, in the Conclusions it is stated that MOFs may provide a route to mesoscale crystalline order, which is difficult to achieve for other systems. This is true, but again there is a vast amount of work on organic single crystals (based on small molecules as well as conjugated oligomers and polymers) which have been grown up to many millimetres in size. The authors should report the size of the crystalline domains in their MOFs, and comment on potential routes to increase domain size towards formation of “crystalline films”. I find e.g. the statement “... extreme significance towards further development ...” too weak, this should be extended to further highlight the broader interest that this work clearly shows.

Response: It is rightly pointed out by the reviewer that the crystalline domain size of these mesoscale ordered SURMOFs is an important parameter that can affect the exciton transport properties. We believe that the exciton diffusion length (~ 97 nm) achieved in the present work can be improved by increasing the crystalline domain size. The crystalline domains in the presented SURMOF range from 50-60 nm. Further development of fabrication technique, such as heteroepitaxial growth of MOFs on metal-hydroxides reported by Falcaro *et al* (*Nat. Mater*, **2017**, 16, 342) can improve the crystalline domain size. In our revised manuscript, we have added a sentence on this and highlighted in yellow.

The sentence “extreme significance towards further development” has been removed.

2. Exciton annihilation (my previous point 2): Although the fluences used in this work are clearly below those, for which annihilation is typically observed. However, the system presented here is clearly not a “typical” system, it is rather optimised for efficient (Figs. 2 & 4) anisotropic energy transport. As such I believe that annihilation sets in already for fluences significantly below “typical” values, i.e., I would like to see fluence-dependent data to verify the absence of annihilation in the present system. In fact, the observation of a significantly lower annihilation threshold would make this work even stronger. Finally, the fluence reported in the authors' reply should be reported in the paper (I could not find this number).

Response: We agree with the reviewer’s comment. Accordingly, we have carried out a new set of measurements and recorded PL spectra in the fluence range of 8.8-380 nJ/cm². These new data (see below) reveal a linear dependence of PL intensity on fluence, suggesting absence of annihilation in this range. Since all measurements on the anisotropic transport were carried out for a maximum fluence of 160 nJ/cm², we can safely conclude that the measured energy transfer efficiencies are unaffected by annihilation effects. A brief discussion of this point has been added to the supporting information of the revised manuscript.

Figure R1: (a) PL spectra with fluence starting from 8.8 nJ/cm² to 380 nJ/cm²; (b) PL intensity plotted against fluence, (Green box= low to high fluence; Red box= high to low fluence).

At much higher fluence values (>1000 nJ/cm²), annihilation processes can become significant. However, in our work such high fluence excitation has not been used.

Following the reviewers' suggestion we have added the fluence values in the revised manuscript.

3. A minor point: Please provide the photon energies used for excitation in Figs. S10, S11, and S14.

Response: We have provided the photon energies in the revised supporting information.

After addressing these issues, this very nice manuscript is to my opinion suited for publication in Nature Communications.

We are grateful to this reviewer for the appreciation of the work. We feel we have adequately accounted for all of his/her criticism.

Reviewer #2 (Remarks to the Author):

This manuscript reports the different migration behaviors of monomer and excimer excited states in the SURMOFs systems. Through time-resolved PL measurements and simulations, the authors concluded that the excited state of the excimer transport is anisotropic, but the monomer excited state transport is less anisotropic. The authors revised their previous submission and responded to comments from previous reviewers, but there are still issues left unaddressed, rendering it unready for publication.

1. SEM images of these samples should be given. The AFM images (Figure S2) showed the roughness of 5 nm for a thin layer of 35 nm in thickness. The sample surface is not smooth at all after comparing the two numbers. Considering the thickness of some samples to be below 20nm, the roughness of 5nm will be considerable and can greatly affect the photophysical measurement. The authors cannot just “trust” literature precedence, as the scientific conclusion of this paper relies on the correct picture more heavily than those in the previous reports.

Response: We have followed the suggestion of the referee and included a SEM image in the revised supporting information.

We agree with the reviewer that the surface roughness of 5 nm corresponding to the height of ~ 2 unit cells will have an influence on the transport of the excited states in thinner films. Such roughness could account for some of the quenching of the excimer emission. However, this would only make the conclusion that excimer transport is highly anisotropic even stronger as the limited quenching observed in the multilayered structures could be due to interfacial effects. In the future, efforts will be directed towards optimizing synthesis conditions in order to obtain SURMOFs with lower roughness.

2. The monomer excited state cannot be 100% isotropic in its transport in an anisotropic structure. The authors should be careful to not use the word isotropic. I think the authors just want to compare the monomer vs. excimer state. If the authors focus on this comparison to even reflect it in the title, the whole paper will be easier to understand. I strongly suggest the authors to completely rewrite the introduction to focus on the directional transport of excimer excited state vs. monomer excited state.

Response: The reviewer is correct. In fact, already in the previous version of the manuscript we have stated that the monomer-related state is not truly isotropic. As suggested by the reviewer, we have reworded the manuscript and have avoided the use of the term “isotropic”.

As suggested by the reviewer, we have reworded our introduction. In the new version, in the second paragraph, we now clearly mention the two different excited states and their different diffusion behavior. We also added a sentence stating that we studied the transport nature of both the excited states and found the excimer state to be anisotropic, while the monomer state can diffuse in other directions also. This part is now highlighted in yellow.

The first paragraph gives an introduction to the general energy transfer phenomenon, importance of long diffusion length with anisotropy, and related materials reported so far (as suggested by reviewer 3 in the previous revision in *Nat. Mater*).

Hence, we believe this introduction is necessary to present our work and it also focus on the work we described in the later part.

3. The difference between the isotropic vs anisotropic model prediction in Figure 4b (black vs. red lines) are very small, while the experimental error can be larger than the difference. This renders the whole discussion and conclusion of this paper unconvincing. I am against the publication of this paper because of this point.

Response: We feel that this criticism of the referee is not justified. All the experimental data points in Figure 4b are clearly below the dotted red line, even after considering the experimental error bar. Hence, we believe that the experimental results truly follow the anisotropic model.

Further, the anisotropy of the excimer state can be easily ascertained from the following experimental evidences:

In the mixed-linker SURMOF-2, the quenching efficiency of the excimer state is higher than the monomer state (Figure 2b). While in the hetero-multilayer SURMOF, the excimer state quenching efficiency is much lower than the monomer state (Figure S21 and S22). This confirms the inefficient diffusion of the excimer state along [100] and [001] direction.

4. I do not know what normalization method is used for Figure S13, but it seems to show that energy transfer is not present, since the acceptor emission intensity does not increase at donor absorption region but only increases at acceptor absorption region when the doping level goes up.

Response: Unfortunately, we do not understand the comment of the reviewer on this point.

The indicated Figure S13 is the excitation spectra of the mixed-linker SURMOFs, where it is shown that monitoring the PL at ~ 2 eV (acceptor emission) we achieve maximum intensity from the donor absorption range (3.2 eV). This observation suggests that the energy is transferred from the donor to the acceptor. As the acceptor concentration increases, we observed a rising intensity ~ 2.5 eV (acceptor absorption). However, the maximum intensity always stayed ~ 3.2 eV suggesting energy transfer process in all the mixed-linker SURMOFs.

5. In the Monte Carlo simulations (Figure S19), the migration of monomer excited state seems also to be anisotropic, why is it not observed in the experiment?

Response: We have already mentioned that the monomer state is not perfectly isotropic. Relative to the excimer state, the monomer state motion along the [001] and [100] direction is efficient. An absolute comparison of quenching efficiencies of the monomer state, in the mixed-linker and hetero-multilayer structures is not possible. Hence the slightly lower diffusion efficiency along [100] and [001] axes, compared to [010] axis cannot be ascertained.

6. A key message the paper should but failed to convey is the reason why excimer excited state transport is more anisotropic than monomer excited state transport. The authors have performed simulations but did not extract the key message from these simulations. The general readers want to know the principles behind it.

Response: As suggested by the reviewer, we have added a discussion on the different transport features of the two excited states in the revised manuscript, and highlighted in yellow.

Reviewer #1 (Remarks to the Author):

I am happy with the revised version of the manuscript by Haldar et al., which, to my opinion, is now suitable for publication.

Reviewer #2 (Remarks to the Author):

The authors have addressed most of the concerns from the reviewers. I think this paper is almost ready for publication. I just want to clarify the issue with Figure S13 and Figure 4.

For Figure S13, my previous comment is mainly about the "normalization method" of Figure S13. I can understand energy transfer does happen in such system, and I feel that Figure S13 must be normalized to the peak of donor excitation, otherwise the data are not self-consistent. The author needs to indicate this normalization method in the figure caption.

For Figure 4, I do not understand why the authors are so positive about the difference between the black line and the red line. The error bar in the figure is about 0.03, while the difference between the black line and the red line is only 0.03-0.05. I believe the authors have plotted their results in other ways to double check their results. But to the readers, Figure 4 is the only entry to support the key points of this paper, and this figure gives an impression of a small difference masked by the experimental error. As the difference of the two models can also lead to differences in other quantities other than the quenching efficiency, I suggest if the authors can also plot the same data in other ways such as the number of donor ligands quenched by one acceptor. Will that show more significant difference? I understand that will be the same data replotted, but the general readers will not believe the result of Figure 4 in its current drawing. At least, the authors need to put the error bar directly on the data points to show that they are definitely not on the red line.

Point-by-point response to the reviewer's comment

Reviewer #1 (Remarks to the Author):

I am happy with the revised version of the manuscript by Halдар et al., which, to my opinion, is now suitable for publication.

Response: We are glad to learn that the reviewer is happy with the modifications and recommends publication of the revised manuscript.

Reviewer #2 (Remarks to the Author):

The authors have addressed most of the concerns from the reviewers. I think this paper is almost ready for publication. I just want to clarify the issue with Figure S13 and Figure 4.

Response: We are glad to know that the reviewer finds this version of the manuscript almost ready for publication. He/she raised some minor questions, which we address in a point-by-point fashion below.

For Figure S13, my previous comment is mainly about the "normalization method" of Figure S13. I can understand energy transfer does happen in such system, and I feel that Figure S13 must be normalized to the peak of donor excitation; otherwise the data are not self-consistent. The author needs to indicate this normalization method in the figure caption.

Response: As suggested by the reviewer, we have added a statement about the normalization method used to the revised manuscript. The excitation spectra of the mixed-linker SURMOFs are all normalized to the donor excitation peak, as recommended by the referee.

For Figure 4, I do not understand why the authors are so positive about the difference between the black line and the red line. The error bar in the figure is about 0.03, while the difference between the black line and the red line is only 0.03-0.05. I believe the authors have plotted their results in other ways to double check their results. But to the readers, Figure 4 is the only entry to support the key points of this paper, and this figure gives an impression of a small difference masked by the experimental error. As the difference of the two models can also lead to differences in other quantities other than the quenching efficiency, I suggest if the authors can also plot the same data in other ways such as the number of donor ligands quenched by one acceptor. Will that show more significant difference? I understand that will be the same data replotted, but the general readers will not believe the result of Figure 4 in its current drawing. At least, the authors need to put the error bar directly on the data points to show that they are definitely not on the red line.

Response: As suggested by the reviewer, we have added the error bars to the data. In our opinion, the comparison clearly reveals that the difference between the black and the red line is outside the error bars.